# Rapid Regulation of Human Mesenchymal Stem Cell Proliferation Using Inducible Caspase-9 Suicide Gene for Safe Cell-Based Therapy

**DOI:** 10.3390/ijms20225759

**Published:** 2019-11-16

**Authors:** Mari Tsujimura, Kosuke Kusamori, Makiya Nishikawa

**Affiliations:** Laboratory of Biopharmaceutics, Faculty of Pharmaceutical Sciences, Tokyo University of Science, 2641 Yamazaki, Noda, Chiba 278-8510, Japan; 3a17704@ed.tus.ac.jp (M.T.); makiya@rs.tus.ac.jp (M.N.)

**Keywords:** inducible caspase-9, AP20187, cell regulation, cell-based therapy

## Abstract

The regulation of transplanted cell proliferation and function is important to achieve safe cell-based therapies. We previously reported that the proliferation and function of transplanted cells, which expressed the *herpes simplex virus thymidine kinase* (*HSVtk*) suicide gene, could be controlled by ganciclovir (GCV) administration. However, there are some concerns regarding the use of GCV. It is reported that the *inducible caspase-9* (*iC9*) gene, a human caspase-9-derived genetically engineered suicide gene, rapidly induces cell apoptosis in the presence of apoptosis inducers, such as AP20187. In this study, we used a combination of the *iC9* gene and AP20187 to achieve rapid regulation of transplanted cell proliferation. Cells from the human mesenchymal stem cell line UE7T-13 were transfected with the *iC9* gene to obtain UE7T-13/iC9 cells. AP20187 significantly reduced the number of UE7T-13/iC9 cells within 24 h in a concentration-dependent manner. This reduction was much faster than the reduction of HSVtk-expressing UE7T-13 cells induced by GCV addition. Subcutaneous AP20187 administration rapidly reduced the luminescence signal from NanoLuc luciferase (Nluc)-expressing UE7T-13/iC9 cells transplanted into mice. These results indicate that the combined use of the *iC9* gene and AP20187 is effective in rapidly regulating transplanted cell proliferation.

## 1. Introduction

Cell-based therapies are excellent treatment methods that involve cell transplantation into patients. In recent years, many cell-based therapies have been developed for the treatment of various diseases including kidney diseases, liver diseases, cardiovascular diseases, and neurological diseases using tissue-specific cells, mesenchymal stem cells (MSCs), and induced pluripotent stem (iPS) cells, or embryonic stem (ES) cell-derived cells. Notably, the development of iPS cells or ES cells opened the possibility of mass production of various kinds of tissue-specific cells, which was one of the biggest hurdles of cell-based therapies for a long time [1,2,3,4,5,6]. In addition to these regenerative therapies, chimeric antigen receptor T-cells are also applied for cancer treatment as a cell-based therapy [7,8,9]. While cell-based therapies have been expected to achieve a high therapeutic effect with a single transplantation compared to conventional drug treatments, there are some problems to be solved in order to establish the safety and effectiveness of cell-based therapies [10,11,12,13].

One concern regarding the use of cells derived from iPS or ES cells is the regulation of transplanted cell proliferation. It is reported that undifferentiated iPS or ES cells infinitely proliferate after transplantation and form teratomas with 80% or higher probability [14]. There are some reports on the differentiation of these stem cells into tissue-specific cells for use in cell-based therapies, but it is difficult to differentiate every cell into the cell of interest. Therefore, there is a risk that some undifferentiated cells may form teratomas. Some attempts have been made to solve this risk [15,16,17], but none have been completely effective.

Another concern is the regulation of transplanted cell function. It was reported that islet transplantation into diabetic patients reduced blood glucose to a normal level that was maintained for a long time after transplantation [18,19]. However, hypoglycemia may occur in islet-transplanted patients in some cases, when islet function is overexpressed by exercise [20]. Therefore, the regulation of transplanted cell function is also highly needed to develop safe cell-based therapies.

We previously succeeded in regulating cell proliferation using *herpes simplex virus thymidine kinase* (*HSVtk*), a suicide gene, and ganciclovir (GCV) based on the theory that HSVtk-expressing cells undergo apoptosis induced by GCV [21]. We demonstrated that the proliferation and function of HSVtk-expressing cells could be regulated by controlling GCV concentration. In addition, a high dose of GCV eliminated the HSVtk-expressing cells transplanted into mice. This system using HSVtk and GCV, however, could have some unsolved problems. First, GCV potentially causes adverse effects, including leukopenia and kidney failure [22,23]. Second, the HSVtk is a viral protein, so it may be immunogenic for humans [24,25]. Third, the elimination of HSVtk-expressing cells by GCV requires several days, because cell apoptosis by HSVtk and GCV is cell cycle-dependent [26]. Therefore, in order to achieve safer and more rapid regulation of transplanted cell proliferation, another system with better properties should be developed.

The *inducible caspase-9* (*iC9*) gene, another suicide gene, is a genetically engineered human caspase-9 [27]. When a chemical inducer of dimerization (CID), such as AP1903 and AP20187, is added to iC9-expressing cells, the iC9 protein becomes dimerized, subsequently directly activates the caspase pathway, and induces cell apoptosis within a few hours [28,29,30,31]. In addition, CID is bioinert, so it should cause few adverse effects. Therefore, the *iC9* gene has recently been applied to some clinical trials, and its usefulness and safety in humans have been demonstrated [32].

Therefore, in this study, we attempted to regulate the proliferation and function of cells transplanted into mice in a short period of time using the *iC9* gene and AP20187. To achieve this, cells from the human mesenchymal stem cell line UE7T-13 were transfected with the *iC9* gene, and UE7T-13/iC9 cells were established. Then, we examined whether AP20187 treatment was able to rapidly regulate the proliferation and function of iC9 gene-expressing cells after transplantation into mice.

## 2. Results

### 2.1. Characteristics of UE7T-13/iC9 Cells

Figure 1 shows the characteristics of the established UE7T-13/iC9 cells. UE7T-13 and UE7T-13/iC9 cells were almost identical in appearance (Figure 1A). To confirm *iC9* gene expression in UE7T-13/iC9 cells, mRNA expression of the *iC9* gene was detected by real-time PCR (Figure 1B) and was found to be high. Western blotting also showed that a strong iC9 band was detected at the position of 47 kDa for the UE7T-13/iC9 cells, but not the UE7T-13 cells (Figure 1C). UE7T-13 and UE7T-13/iC9 cells showed a comparable ability to proliferate and differentiate to adipocytes or osteoblasts (Figure 1D,E).

### 2.2. Sensitivity of UE7T-13/iC9 Cells to AP20187 and UE7T-13/HSVtk Cells to GCV

When UE7T-13 and UE7T-13/iC9 cells were cultured in medium containing AP20187, UE7T-13/iC9 cell viability decreased as AP20187 concentration increased, whereas UE7T-13 cell viability hardly changed (Figure 2A). In addition, most UE7T-13/iC9 cells died by 6 h after AP20187 addition at 0.05 nM or higher. Furthermore, when UE7T-13 and UE7T-13/HSVtk cells were cultured in GCV-containing medium, the viability of UE7T-13/HSVtk cells, but not that of UE7T-13, decreased with increasing GCV concentration (Figure 2B). UE7T-13/HSVtk cells required 96 h or longer to die at GCV concentrations of 500 nM or less.

### 2.3. Effect of AP20187 on the Proliferation of UE7T-13/iC9 and UE7T-13/iC9/Nluc Cells

To examine the regulation of iC9-expressing cell proliferation, UE7T-13/iC9 cells were cultured in culture media containing various concentrations of AP20187, and the cell number was measured every two days (Figure 3A,B). The number of UE7T-13/iC9 cells in the AP20187-containing media decreased in an AP20187 concentration-dependent manner. On the other hand, the number of UE7T-13/iC9 cells in AP20187-free medium increased with time. In addition, we confirmed that the cell number repeatedly increased and decreased depending on the presence or absence of AP20187 (Appendix A). Similarly, UE7T-13/iC9/Nluc cells were cultured and the luciferase activity in the supernatant was measured (Figure 3C,D). The luciferase activity of UE7T-13/Nluc cells increased with time irrespective of the presence or absence of AP20187. Conversely, the luciferase activity of UE7T-13/iC9/Nluc cells decreased in an AP20187 concentration-dependent manner.

### 2.4. Elimination of UE7T-13/iC9/Nluc Cells by AP20187 and UE7T-13/HSVtk/Nluc Cells by GCV

Figure 4 shows the luminescence images of mice transplanted with UE7T-13/iC9/Nluc or UE7T-13/HSVtk/Nluc cells before and after AP20187 or GCV administration, respectively. In Figure 4A, the upper images showed that cell-derived luminescence was detected on day 2 without AP20187 administration. On the other hand, the lower images showed that cell-derived luminescence was not detected on day 2 with AP20187 administration. In Figure 4B, the upper images showed that cell-derived luminescence was detected on day 6 without GCV administration. On the other hand, the lower images showed that cell-derived luminescence was not detected on day 6 with GCV administration.

## 3. Discussion

It is important to regulate the proliferation and function of transplanted cells to achieve optimal cell-based therapies with high safety. It has been previously reported that undifferentiated ES cells can be removed using the monoclonal antibody mAB 84 and anti-stage-specific embryonic antigen-5 [16,17]. However, it is difficult to completely remove undifferentiated cells using this method. Another approach has also been reported, wherein the Tet-on/off system regulates transplanted cell function following the administration of doxycycline or related compounds [33,34,35,36]. The Tet-on or Tet-off system requires continuous doxycycline administration to express or stop the function of the transduced gene in the cells, which decreases the quality of life of patients. The combination of the *iC9* gene and CID, including AP20187, can precisely regulate the number of transplanted cells by controlling the CID dose, and cell function can also be regulated simultaneously. Therefore, CID administration is required only when transplanted cells form teratomas or tumors, or when their function exceeds the desired level. This means that continuous CID administration is not required, unlike that in the Tet-on/off system.

It has been previously reported that HSVtk-expressing cells induced apoptosis four to five days after GCV administration [26,37]. Consistent with these reports, our previous study also showed that HSVtk-expressing cells required two to four days in vitro and three days in vivo, respectively, to induce apoptotic cell death. In contrast, iC9-expressing cell apoptosis has been reported to be induced within a few hours after adding the apoptosis inducer. This is because iC9 directly activates the apoptotic pathway [28,29,30,31]. The results of the present study also showed that most iC9-expressing cells died within 6 h after adding AP20187 (Figure 2A). Moreover, we succeeded in demonstrating that the proliferation and function of iC9-expressing cells could be rapidly and strictly regulated by controlling the concentration of AP20187 (Figure 3B). This rapid regulation cannot be achieved with the *HSVtk* gene and GCV. Therefore, we can select these methods to regulate cell proliferation depending on the situation, and the *iC9* gene is especially useful for rapid cell regulation.

The product of the *HSVtk* gene, HSVtk protein, can be immunogenic and may cause allergic symptoms or the rejection of transduced cells by the host immune system [24,38,39]. In contrast, the iC9 protein is a human protein, so iC9-expressing cells may be minimally rejected or cause few allergic symptoms. Apoptosis inducers should be safe for clinical use. Regarding this point, the combination of the *iC9* gene and AP20187 can overcome some hurdles, including the immunogenicity and adverse effects of apoptosis inducers; thus, the cell regulation system using the *iC9* gene and AP20187 may be safe for human use.

CIDs including AP1903 and AP20187 bind to iC9 protein and induce cell apoptosis, and these chemicals were synthesized based on the structure of the chemical FK1012. AP20187 is a derivative of AP1903 and was designed to improve the pharmacological properties of AP1903 [40]. Because these chemicals can achieve effective induction of cell apoptosis in iC9-expressing cells at low concentrations, they have been used often in recent studies. In this study, we selected AP20187, which is a more effective apoptosis inducer than AP1903, as the CID. It has been reported that the elimination half-life of AP20187 is 7.24 h in healthy juvenile baboons [41]. In this study, 3.7 μg/kg AP20187 was subcutaneously administered twice at a 12 h interval. Further studies regarding the administration route and pharmacokinetics of AP20187 are required to effectively regulate iC9-expressing cells in vivo.

## 4. Materials and Methods

### 4.1. Animals

Male BALB/c Slc-nu/nu mice (5 weeks old) were purchased from Japan SLC, Inc. (Shizuoka, Japan) and maintained under specific pathogen-free conditions. The protocols for experiments involving animals were approved by the Institutional Animal Experimentation Committee of Tokyo University of Science (Y19036; 31 May, 2019). All experiments involving animals were conducted in accordance with the procedures outlined in the National Institutes of Health Guide for the Care and Use of Laboratory Animals.

### 4.2. Materials

The pMSCV-F-del Casp9.IRES.GFP plasmid was obtained from Addgene (Cambridge, MA, USA). The pSelect-zeo-HSV1tk plasmid, LB broth base, and LB agar were purchased from InvivoGen Co. (San Diego, CA, USA). The pNL2.3 [secNluc/Hygro] plasmid was obtained from Promega Co. (Tokyo, Japan). KOD-Plus-Neo was purchased from TOYOBO Co., Ltd. (Osaka, Japan). Competent NEB 10-beta *Escherichia coli* (high efficiency), EcoRI-HF, and NotI-HF were purchased from New England Biolabs Inc. (Ipswich, MA, USA). XbaI, NotI, B/B Homodimerizer (AP20187), pLVSIN-CMV-Neo plasmid, and Lentiviral High Titer Packaging Mix were purchased from Takara Bio Inc. (Shiga, Japan). HEPES was purchased from Dojindo Laboratories (Kumamoto, Japan). Ganciclovir hydrate and forskolin were purchased from Tokyo Kasei Kogyo Co., Ltd. (Tokyo, Japan). Hank’s balanced salt solution was purchased from Sigma-Aldrich (St. Louis, MO, USA). Alizalin Red S, NP-40 substrate, tris(hydroxymethyl)aminomethane, sodium deoxycholate, sodium dodecyl sulfate, chloroform, isopropyl alcohol, ethanol, NaHCO_3_, D-(+)-glucose, and 0.4 *w*/*v* % trypan blue solution were purchased from FUJIFILM Wako Pure Chemical Co. (Osaka, Japan). Fetal bovine serum (FBS) was purchased from Thermo Fisher Scientific (Waltham, MA, USA). Dulbecco’s modified Eagle’s medium (DMEM) was purchased from Nissui Seiyaku (Tokyo, Japan). Oil Red O, polyethylene glycol #6000, penicillin-streptomycin-glutamine mixed solution, and 100 mM sodium pyruvate solution (100 ×) were purchased from Nacalai Tesque, Inc. (Kyoto, Japan). All other chemicals used were of the commercially available highest grade.

### 4.3. Cell Culture

UE7T-13 human mesenchymal stem cells were obtained from the JCRB Cell Bank (Osaka, Japan). UE7T-13 cells were cultured in DMEM supplemented with 10% heat-inactivated FBS and penicillin-streptomycin-glutamine mixed solution at 37 °C in humidified air containing 5% CO_2_. Lenti-X 293T cells were purchased from Takara Bio Inc. (Shiga, Japan) and cultured in DMEM supplemented with 10% heat-inactivated FBS, penicillin-streptomycin-glutamine mixed solution, and 1 mM sodium pyruvate solution (100 ×) at 37 °C in humidified air containing 5% CO_2_.

### 4.4. Construction of Plasmids

The primer sequences used to construct plasmids are summarized in Table 1. To construct an iC9-encoding plasmid, pLVSIN-iC9, the cDNA fragment of the *iC9* gene was amplified via polymerase chain reaction (PCR) using primers from the pMSCV-F-del Casp9.IRES.GFP plasmid. Subsequently, the amplified PCR product and pLVSIN-CMV-Neo plasmid were digested using XbaI and NotI restriction enzymes overnight at 37 °C. Ligation and cloning in *E. coli* were performed as previously reported [17]. To construct an *HSVtk*-encoding plasmid, pLVSIN-HSVtk, the cDNA fragment of the *HSVtk* gene was amplified via PCR using primers from the pSelect-zeo-HSV1tk plasmid. Subsequently, the amplified PCR product and pLVSIN-CMV-Neo vector were digested using EcoRI-HF and NotI-HF restriction enzymes overnight at 37 °C. Ligation and cloning were performed as described above. Furthermore, to construct an Nluc-encoding plasmid, pLVSIN-Nluc, the cDNA fragment of the Nluc gene was amplified via PCR using primers from the pNL2.3 [secNluc/Hygro] plasmid. Subsequently, the amplified PCR product and pLVSIN-CMV-Neo plasmid were digested using XbaI and NotI restriction enzymes overnight at 37 °C. Ligation and cloning were performed as described above.

### 4.5. Establishment of Cells

Lenti-X 293T cells (1 × 10^7^ cells) were seeded in a 10 cm culture dish and cultured overnight at 37 °C in humidified air containing 5% CO_2_. Lenti-X 293T cells were co-transfected with the pLVSIN-iC9, pLVSIN-HSVtk, or pLVSIN-Nluc plasmid and the Lentiviral Packaging Mix by the calcium phosphate method. After 24 h, the medium was replaced with culture medium containing 10 μM forskolin, and cells were cultured for an additional 48 h. Then, the supernatant was collected and centrifuged at 900× *g* for 5 min. Furthermore, the supernatant was filtered, mixed with PEG solution (32 *w*/*v* % polyethylene glycol #6000, 400 mM NaCl, and 40 mM HEPES), and centrifuged at 2500× *g* for 40 min. The precipitated viruses were collected by Opti-MEM and transduced into UE7T-13, UE7T-13/iC9, or UE7T-13/HSVtk cells. After 24 h, the medium was replaced, and the cells were cultured in normal culture medium for a few days. These cells were cloned and incubated until they were confluent. Cloned cells, which underwent apoptosis by AP20187 or GCV, were selected as UE7T-13/iC9 or UE7T-13/HSVtk, respectively. Nluc expression in UE7T-13/Nluc, UE7T-13/iC9/NLuc, and UE7T-13/HSVtk/Nluc cells was confirmed by detecting luminescence using a Nano-Glo assay reagent (Promega Co., Tokyo, Japan).

### 4.6. The mRNA Expression of the iC9 Transgene

RNA was extracted from UE7T-13 and UE7T-13/iC9 cells using Sepasol RNA I Super G (Nacalai Tesque, Inc.). The extracted RNA was converted to cDNA using ReverTra Ace qPCR RT Master Mix (TOYOBO Co., Ltd.). The real-time PCR reaction was performed using the CFX Connect Real-Time System (Bio-rad, Hercules, CA, USA), THUNDERBIRD SYBR qPCR Mix (TOYOBO Co., Ltd.), and specific primers. Primer sequences are shown in Table 1. The cycle was conducted at 95 °C for 30 sec, followed by PCR amplification with 40 cycles of 95 °C for 5 sec, 55 °C for 15 sec, and 72 °C for 45 sec. The gene expression rate was calculated by the 2^–ΔΔC^_T_ method [42].

### 4.7. Western Blotting 

UE7T-13 or UE7T-13/iC9 cells were suspended in RIPA buffer (50 mM Tris-HCl, 150 mM NaCl, 0.5 *w*/*v* % sodium deoxycholate, 0.1 *w*/*v* % sodium dodecyl sulfate, and 1 *w*/*v* % NP-40 substitute) and incubated for 30 min on ice to obtain the proteins. After 20 min centrifugation, the supernatant was collected and heated to 95 °C for 3 min with 1 M dithiothreitol and 4× SDS-PAGE sample buffer. Then, 30 μg total protein was separated by SDS-PAGE (Bio-Rad, Berkley, CA, USA) and then transferred to polyvinylidene difluoride membranes (Merck Millipore, Stockholm, Sweden). The membranes were incubated with primary antibodies against caspase-9 (#9502; Cell Signaling Technologies, Danvers, MA, USA) followed by goat anti-rabbit horseradish peroxidase-conjugated secondary antibodies (#7074; Cell Signaling Technologies, Beverly, MA, USA). These proteins were detected using Immobilon Western Chemiluminescent HRP Substrate (Millipore Corporation, Billerica, MA, USA) and the LAS-4000 mini imaging system (Fujifilm Co., Ltd., Tokyo, Japan).

### 4.8. Cell Proliferation

UE7T-13 or UE7T-13/iC9 cells (5 × 10^4^ cells) were seeded in a 6-well culture plate, and the cell number was counted every 24 h for 5 consecutive days using the trypan blue dye exclusion method [43].

### 4.9. Cell Differentiation

For differentiation into adipocytes, UE7T-13 or UE7T-13/iC9 cells (5 × 10^3^ cells) were seeded in a 96-well culture plate and cultured for 3 days at 37 °C in humidified air containing 5% CO_2_. Then, the medium was replaced with mesenchymal stem cell adipogenic differentiation medium (Promocell GmbH, Heidelberg, Germany) for 22 days and refreshed every 3 days. The cells that were induced to differentiate into adipocytes were stained using Oil Red O for adipocyte detection. Similarly, for differentiation into osteoblasts, UE7T-13 or UE7T-13/iC9 cells (5 × 10^3^ cells) were seeded in a 96-well culture plate and cultured for 5 days at 37 °C in humidified air containing 5% CO_2_. Then, the medium was replaced with mesenchymal stem cell osteogenic differentiation medium (Promocell GmbH) for 21 days and refreshed every 3 days. The cells that were induced to differentiate into osteoblasts were stained using Alizarin Red S for osteoblast detection. These stained cells were examined under a BZ-9000 digital microscope (Keyence, Osaka, Japan).

### 4.10. Sensitivity of UE7T-13/iC9 Cells to AP20187 and UE7T-13/HSVtk Cells to GCV

UE7T-13/iC9 or UE7T-13/HSVtk cells (1 × 10^4^ cells) were seeded in a 96-well culture plate and cultured overnight at 37 °C in humidified air containing 5% CO_2_. The next day, the medium was replaced with fresh medium containing various concentrations of AP20187 or GCV, and the cell number was evaluated by Cell Counting Kit-8 (Dojindo Laboratories) for 6, 24, 48, and 96 h after adding AP20187 or GCV. The cell number of the no-treatment group at 6 h was set as 100%.

### 4.11. In Vitro Regulation of UE7T-13/iC9 Cell Proliferation

To evaluate the regulation of UE7T-13/iC9 cell proliferation depending on AP20187 concentration, UE7T-13 or UE7T-13/iC9 cells (5 × 10^3^ cells) were seeded in a 96-well culture plate and cultured for 24 h at 37 °C in humidified air containing 5% CO_2_, and the cell number was evaluated by Cell Counting Kit-8. The cells were cultured in medium containing various AP20187 concentrations, and the cell number was evaluated every 2 days using Cell Counting Kit-8. The cell number was calculated by setting the cell number on day 0 as 100%. Furthermore, UE7T-13/iC9 cells (5 × 10^3^ cells) were seeded in a 96-well culture plate and cultured for 24 h at 37 °C in humidified air containing 5 % CO_2_, and the cell number was evaluated by Cell Counting Kit-8. The cells were cultured in medium containing 0.05 nM AP20187 initially for two days (day 0 to day 2). Then, these cells were cultured in normal medium for four days (day 2 to day 6). Finally, these cells were cultured in 0.05 nM AP20187 for two days (day 6 to day 8). The cell number was evaluated every 2 days using Cell Counting Kit-8 and calculated by setting the cell number on day 0 as 100%.

UE7T-13/Nluc or UE7T-13/iC9/Nluc cells (1 × 10^3^ cells) were seeded in a 96-well culture plate and cultured for 48 h at 37 °C in humidified air containing 5% CO_2_. The medium was collected and replaced with fresh medium containing various AP20187 concentrations every 2 days. The cellular function was evaluated by measuring the Nluc activity, which was calculated by setting Nluc activity on day 0 as 100%.

### 4.12. Elimination of UE7T-13/iC9/Nluc Cells and UE7T-13/HSVtk/Nluc Cells in Mice

UE7T-13/iC9/Nluc or UE7T-13/HSVtk/Nluc cells (1 × 10^6^ cells) were transplanted into the back of BALB/c Slc-nu/nu mice. At 24 h after cell transplantation, 3.7 μg/kg AP20187 was subcutaneously administered twice at a 12 h interval to mice transplanted with UE7T-13/iC9/Nluc cells, or 25 mg/kg GCV was subcutaneously administered every 12 h for 5 days to mice transplanted with UE7T-13/HSVtk/Nluc cells. Cell luminescence in mice was detected using the In-Vivo Xtreme imaging system (Bruker, Billerica, MA, USA) after injecting the Nano-Glo assay reagent into the cell transplantation site at a dose of 50 μL/mouse.

### 4.13. Statistical Analysis

Statistical differences were evaluated by one-way analysis of variance, followed by the Dunnett’s test for multiple comparisons.

## 5. Conclusions

The proliferation of iC9-expressing UE7T-13 cells was rapidly regulated by AP20187 both in vitro and in vivo, indicating that the combination of the iC9 gene and AP20187 is useful to achieve safe cell-based therapies.

## Figures and Tables

**Figure 1 ijms-20-05759-f001:**
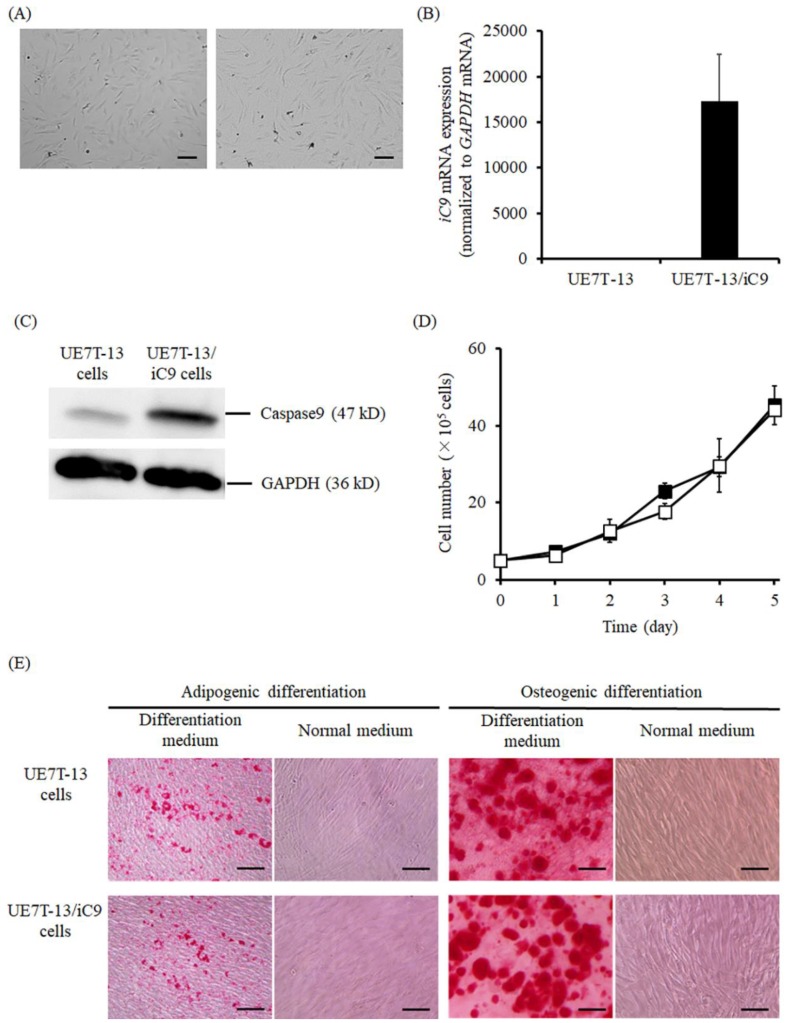
Characteristics of UE7T-13/iC9 cells. (**A**) Typical images of UE7T-13 and UE7T-13/iC9 cells. Scale bars represent 100 μm. (**B**) The mRNA expression of the *inducible caspase-9* (*iC9*) gene. (**C**) Caspase-9 expression in UE7T-13 and UE7T-13/iC9 cells. (**D**) Proliferation of UE7T-13 and UE7T-13/iC9 cells. UE7T-13 cells (white square) and UE7T-13/iC9 cells (black square) are indicated. Results are expressed as the mean ± SD of four samples. (**E**) Differentiation of UE7T-13 and UE7T-13/iC9 cells. Typical images of UE7T-13 and UE7T-13/iC9 cells that differentiated into adipocytes or osteoblasts. Scale bars represent 50 μm.

**Figure 2 ijms-20-05759-f002:**
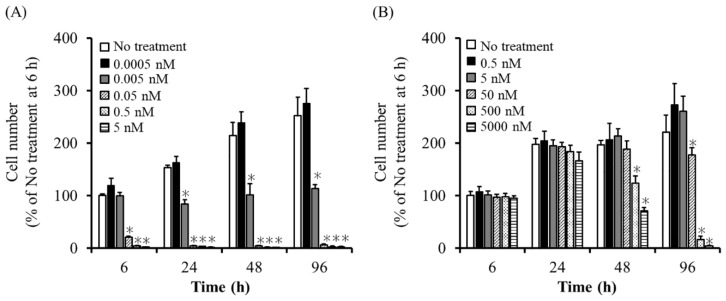
Sensitivity of UE7T-13/iC9 cells or UE7T-13/HSVtk cells to AP20178 or ganciclovir (GCV), respectively. (**A**) The viability of UE7T-13/iC9 cells cultured with AP20187 at various concentrations. Cells were cultured in medium containing various AP20187 concentrations. Results are expressed as the mean ± SD of three to four samples. A representative of three independent experiments with similar results is shown. * *p* < 0.05; statistically significant differences observed in comparison with the no-treatment group. (**B**) The viability of UE7T-13/HSVtk cells cultured with GCV at various concentrations. Cells were cultured in medium containing various GCV concentrations. Results are expressed as the mean ± SD of four samples. A representative of three independent experiments with similar results is shown. * *p* < 0.05; statistically significant differences observed in comparison with the no-treatment group.

**Figure 3 ijms-20-05759-f003:**
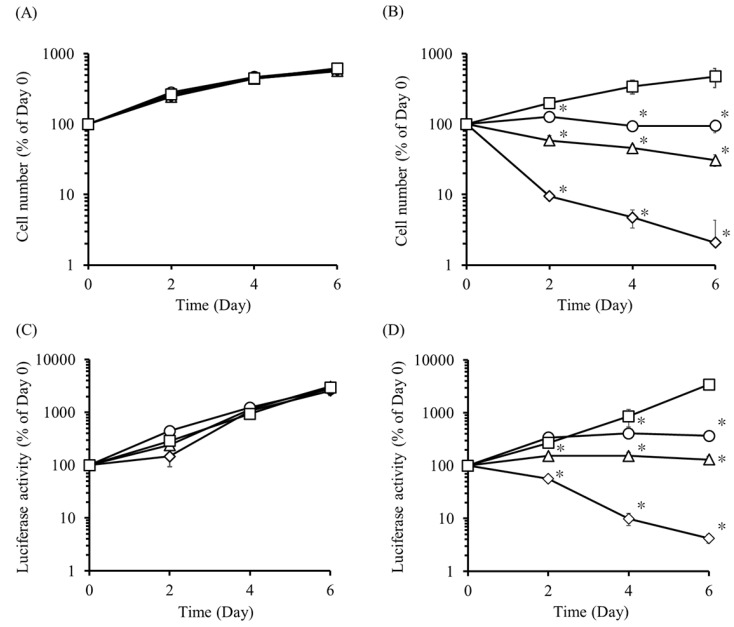
Effect of AP20187 on the proliferation of UE7T-13/iC9 and UE7T-13/iC9/Nluc cells. The number of (**A**) UE7T-13 cells or (**B**) UE7T-13/iC9 cells in media containing 10, 20, or 50 pM AP20187. No treatment (white square), 10 pM AP20187 (white circle), 20 pM AP20187 (white triangle), and 50 pM AP20187 (white diamond) are indicated. Results are expressed as the mean ± SD of four samples. A representative of three independent experiments with similar results is shown. * *p* < 0.05; statistically significant differences observed in comparison with the no-treatment group. The luciferase activity in the supernatant of (**C**) UE7T-13/Nluc and (**D**) UE7T-13/iC9/Nluc cells cultured in normal medium or medium containing 10, 20, or 50 pM AP20187. The luciferase activity was measured every 48 h. No treatment (white square), 10 pM AP20187 (white circle), 20 pM AP20187 (white triangle), or 50 pM AP20187 (white diamond) are indicated. Results are expressed as the mean ± SD of four samples. A representative of three independent experiments with similar results is shown. * *p* < 0.05; statistically significant differences observed in comparison with the no-treatment group.

**Figure 4 ijms-20-05759-f004:**
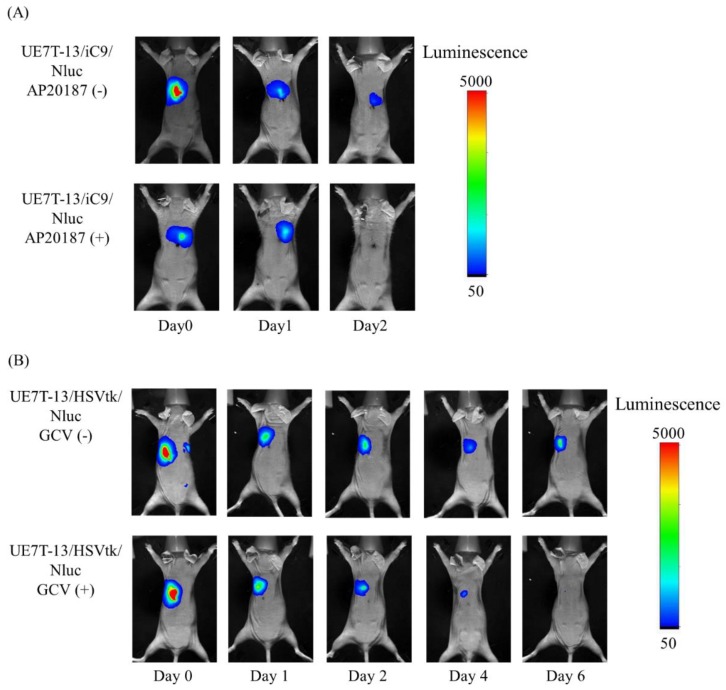
Elimination of UE7T-13/iC9/Nluc or UE7T-13/HSVtk/Nluc cells from mice by AP20187 or GCV, respectively. UE7T-13/iC9/Nluc (**A**) or UE7T-13/HSVtk/Nluc (**B**) cells were subcutaneously transplanted into the back of BALB/c Slc-nu/nu mice. AP20187 (3.7 μg/kg) was subcutaneously administered twice to UE7T-13/iC9/Nluc cell-transplanted mice at a 12 h interval, and GCV (25 mg/kg) was subcutaneously administered to UE7T-13/HSVtk/Nluc cell-transplanted mice every 12 h for 5 days. The luminescence of the cells transplanted into mice was detected in an In-Vivo Xtreme imaging system. (−); without AP20187 or GCV administration, (+); with AP20187 or GCV administration.

**Table 1 ijms-20-05759-t001:** Primers.

**Primers for Construction of Plasmids**	
***iC9***	
Forward	5′- TGCTCTAGAATGCTCGAGGGAGTGC -3′
Reverse	5′- TAAAGCGGCCGCTTAGTCGAGTGCG -3′
*HSVtk*	
Forward	5′- CCGGAATTCATGGCTTCTTACCCTG -3′
Reverse	5′- TAAAGCGGCCGCTTAGTTGGCCTCT -3′
*Nluc*	
Forward	5′- TGCTCTAGAATGAACTCCTTCTCCACAAG -3′
Reverse	5′- TAAAGCGGCCGCTTACGCCAGAATGCGTT -3′
**Primers for RT-PCR**	
*GAPDH*	
Forward	5′- GCACCGTCAAGGCTGAGAAC -3′
Reverse	5′- ATGGTGGTGAAGACGCCAGT -3′
*iC9*	
Forward	5′- TGTGGGTCAGAGAGCCAAAC-3′
Reverse	5′- CAAATCTGCATTTCCCCTCA -3′
*HSVtk*	
Forward	5′- AACATCTACACCACCCAGCAC -3′
Reverse	5′- GAACAGCATCAGTCACAGCATAG -3′

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
