# Peer review of "Rapid Regulation of Human Mesenchymal Stem Cell Proliferation Using Inducible Caspase-9 Suicide Gene for Safe Cell-Based Therapy"

_ijms, 2019, doi:10.3390/ijms20225759_

Round 1
Reviewer 1 Report
The authors submitted very interesting paper focused on proliferation regulation of MSCs by using inducible caspase-9 suicide gene. The article is written on an excellent level. Authors used only relevant methods and obtained results were well illustrated and discussed. I would like to ask why they did not include information about chondrogenic differentiation of MSCs. According to Internatiobal Society for Cellular Therapy MSCs are characterised by three lineage differentiation. I suggest that the authors add evidence about chondrogenic differentiation in characterization of MSCs .
Author Response
Response to Reviewer 1 Comments
First, we would like to express our deepest thanks and appreciation to the reviewers for raising important issues and providing us helpful suggestions. We have extensively revised our manuscript in the light of the issues raised by the reviewers.
Point 1: The authors submitted very interesting paper focused on proliferation regulation of MSCs by using inducible caspase-9 suicide gene. The article is written on an excellent level. Authors used only relevant methods and obtained results were well illustrated and discussed. I would like to ask why they did not include information about chondrogenic differentiation of MSCs. According to Internatiobal Society for Cellular Therapy MSCs are characterised by three lineage differentiation. I suggest that the authors add evidence about chondrogenic differentiation in characterization of MSCs
Response 1: We appreciate the reviewer’s thoughtful comment. As the reviewer pointed out, it is important to evaluate the differentiation of MSCs into three cell lineages: adipocytes, osteoblasts, and chondrocytes. However, in some articles on MSCs, the differentiation into only adipocytes and osteoblasts, two of the three cell lineages, was evaluated *1,*2. The aim of this study is the regulation of the transplanted cell proliferation using the combination of iC9 gene and AP20187. The differentiation of UE7T-13 cells into adipocytes and osteoblasts was evaluated to confirm whether the iC9 gene transfer hardly changed the properties of the cells. Based on the above considerations, we have not evaluated the differentiation of UE7T-13 cells into chondrocytes.
*1: Jaiswal, RK.; Jaiswal, N.; Bruder, SP.; Mbalaviele, G.; Marshak, DR,; Pittenger, MF. Adult human mesenchymal stem cell differentiation to the osteogenic or adipogenic lineage is regulated by mitogen-activated protein kinase. J Biol Chem 2000, 275, 9645-52.
*2: Han, I.; Kwon, BS.; Park, HK.; Kim, KS. Differentiation potential of mesenchymal stem cells is related to their intrinsic mechanical properties. Int Neurourol J 2017, 21, S24-31.
The manuscript underwent English language editing by Editage (JOB CODE: QEXHY_10).
We sincerely hope that you will find these revisions and corrections satisfactory and that our manuscript will now be acceptable for publication in International Journal of Molecular Sciences.
Reviewer 2 Report
The manuscript entitled “Rapid regulation of human mesenchymal stem cell proliferation using inducible caspase-9 suicide gene for safe cell-based therapy” describes a new approach for the control of stem cell proliferation through the overexpression and chemical modulation of inducible caspase-9. This new approach seems to present more advantages than the previous one proposed by the same authors (Tsujimira et al., J Control Release, 2018, 275, 78-84) and so it could represent an useful tool for the amelioration of the stem cell transplant approaches. The manuscript is well written and the results are promptly presented, however I have some concerns that need to be addressed in order to consider the paper for publication.
Introduction:
I think that the first sentence of introduction is too optimistic. I agree with the author that cell-based therapies, in theory, could be better than conventional drug treatment. However, nowadays, there are still many issues that need to be elucidated before the routinely application of stem-cell therapy in clinical practice. I suggest the authors to re-formulate the sentence and to add some references that describe the studies conducted on stem cell therapy in different pathologies with pros and cons: (Wang et al. (PMID 30989038), Ghiroldi et al. (PMID 30332812), Volkman et al. (PMID 28589621), Rota et al. (PMID 31181604).
Results:
Paragraph 2.1 – lane 31: the authors stated that the iC9 band was not detectable in UE7T-13 cells, however in fig 1C the band at 47kDa is detected in both samples. Please, correct. Figure 1: if possible, I suggest the author to increase the quality of the phase-contrast images (1A and 1E). Moreover, in figure 1C I think that it would be better to substitute the lane name (a) and (b) with the cell line name UE7T-13 and UE7T-13/iC9, respectively. Figure 2: even if the symbol explanation is present in the figure caption, I think that the addition of the legend directly in the graph, will increase its understanding. Moreover, I don’t think that this type of graph is the best way to represent these data. This graph should be used to represent data in function of time, instead the figure reported data in function of different AP20187 treatment concentrations. I think that an histogram will be clearer. Paragraph 2.4 – lane 18-19. The sentence “In contrast, (…) after AP20187”, seems in contrast with the previous one and I don’t understand to which figure is referred. Please, explain.
Discussion
The approach proposed in this paper aims to kill the undifferentiated cells after cell transplant to reduce the risk of teratoma formation. However, which is the author’s opinion, regarding the possible effect of AP20187 treatment on the differentiated cells that have to exert their therapeutical function? There could be the risk that AP20187 treatment could reduce their number and thus their effect?
Reviewer 3 Report
Great original article.
As a recommendation in the discussion, it is necessary to describe in detail the mechanisms by which cells not modified by caspase-9 can avoid induced apoptosis.
Author Response
Response to Reviewer 3 Comments
First, we would like to express our deepest thanks and appreciation to the reviewers for raising important issues and providing us helpful suggestions. We have extensively revised our manuscript in the light of the issues raised by the reviewers.
Point 1: Great original article. As a recommendation in the discussion, it is necessary to describe in detail the mechanisms by which cells not modified by caspase-9 can avoid induced apoptosis.
Response 1: We would like to apologize for the ambiguous description about the mechanism of cell suicide induced by iC9 gene and AP20187. First, we transfected UE7T-13 cells with the iC9 gene, and these cells were cloned as UE7T-13/iC9 cells. Therefore, all the UE7T-13 cells stably expressed the iC9 gene. Upon addition of AP20187, iC9 protein in the iC9-expressing cells forms dimer and the caspase pathway is subsequently activated. As a result, the iC9-expressing cells undergo cell death by the addition of AP20187. On the other hand, since the caspase pathway of iC9 non-expressing cells is not activated by the addition of AP20187, AP20187 does not affect the proliferation and survival of iC9 non-expressing cells.
The manuscript underwent English language editing by Editage (JOB CODE: QEXHY_10).
We sincerely hope that you will find these revisions and corrections satisfactory and that our manuscript will now be acceptable for publication in International Journal of Molecular Sciences.
Reviewer 4 Report
In this paper, Tsujimura et al., are using the UE7T-13 human mesenchymal stem cell line engineered to express the inducible caspase-9 (iC9) gene to draw their conclusions. My understanding of the data is that iC9 can be used to trigger apoptosis in cells expressing it following transplantation in vivo thus it could be used as a safety switch in stem cell therapies. Unfortunately this is already been shown before when the iC9 was engineered and the authors cite several of these papers. In addition, authors identify several specific problems in stem cell treatments: (1) that undifferentiated stem cells make teratomas when transplanted, (2) the function of those may not be ideal (like when islets are used). The iC9 system/method that they describe does not help in any of these instances since when AP20187 is administered, all cells with go apoptotic (including the correctly differentiated ones) and they do not show any functional data of cellular activity when lower concentrations than the IC50 are used. Taken together, the paper lacks novelty since the data do not appear to provide any additional information to the already published literature other than that the UE7T-13 cell line undergoes apoptosis in the presence of iC9/AP20187.
